# Effects of bouncing the barbell in bench press on throwing velocity and strength among handball players

**Jørund Løken**, **Tom Erik Jorung Solstad‡**, **Nicolay Stien‡**, **Vidar Andersen‡**, **Atle Hole Saeterbakken**(ID)*

Faculty of Education, Arts and Sports, Western Norway University of Applied Sciences, Sogndal, Norway

☯ These authors contributed equally to this work.
‡ TEJS, NS and VA also contributed equally to this work.
* atle.saeterbakken@hvl.no

**Data Availability Statement:** All relevant data are within the manuscript and its Supporting Information files.

## Abstract

Bench press is a popular training-exercise in throw related sports such as javelin, baseball and handball. Athletes in these sports often use bouncing (i.e., letting the barbell collide with the chest) to create an increased momentum to accelerate the barbell upwards before completing the movement by throwing the barbell. Importantly, the effects of the bouncing technique in bench press have not been examined. Therefore, the aim of this study was to compare the effects of bench press throw with ($BPT_{bounce}$) or without bounce (BPT) on throwing velocity (penalty and 3-step), 1-repetition maximum (1-RM) and average power output (20-60kg) in bench press among handball players. Sixteen male amateur handball players (7.1±1.9 years of handball experience) were randomly allocated to an eight-week supplementary power training program (2 x week$^{-1}$) with either the BPT or $BPT_{bounce}$. Except for the bounce technique, the training programs were identical and consisted of 3 sets with 3–5 repetitions at 40–60% of 1-RM with maximal effort in free-weight barbell bench press throw. The results revealed no significant differences between the groups in any of the tests (p = 0.109–0.957). However, both groups improved penalty throw (BPT; 4.6%, p<0.001, ES = 0.57; $BPT_{bounce}$; 5.1%, p = 0.008, ES = 0.91) and 1-RM (BPT; 9.7%, p<0.001, ES = 0.49; $BPT_{bounce}$; 8.7%, p = 0.018, ES = 0.60), but only the BPT improved the 3-step throw (BPT; 2.9%, p = 0.060, ES = 0.38; $BPT_{bounce}$; 2.3%, p = 0.216, ES = 0.40). The BPT improved power output only at 20kg and 30kg loads (9.1% and 12.7%; p = 0.018–0.048, ES = 0.43–0.51) whereas $BPT_{bounce}$ demonstrated no significant differences across the loads (p = 0.252–0.806). In conclusion, the bounce technique demonstrated similar effects on throwing velocity, muscle strength and muscle power output as conventional bench press throw without the bounce technique.

## Introduction

Coaches and researchers have developed resistance training programs with the intent to maximize the transfer of training-effects to high-velocity movements (i.e., sprinting, kicking,

**Funding:** The funders had no role in study design, data collection and analysis, decision to publish, or preparation of the manuscript. The authors received no specific founding for this work. As of this, a) there are no sources of funding (financial or material support) for this study to report. Furthermore, b) there were no founders involved and therefore the funders had no role in study design, data collection and analysis, decision to publish, or preparation of the manuscript. Finally, c) none of authors received any form of salary form any founders. To clarify everything: The authors received no specific funding for this work.

**Competing interests:** The authors have declared that no competing interests exist.

jumping) [1, 2]. In sports involving overarm throws (e.g., handball, javelin, water polo, baseball), maximizing release velocity is one of the most critical parameters for success [3]. Previous studies have demonstrated that throwing velocity may be increased through various training methods (i.e., traditional strength training, power training, core stability training, and throwing with underweight- and overweight balls) [4–7]. Potentially, these various resistance training approaches may result in different adaptations [8, 9], and different effects on throwing performance [10, 11].

In sports relying on the overarm throw, the barbell bench press exercise is one of the most utilized exercises for developing upper body strength and power [4, 6, 10, 12]. The bench press movement can be altered using various equipment, intensities, and lifting techniques [9, 13]. However, it is not yet clear which resistance training approach that maximize performance in high-velocity movements such as the overarm throw [2, 4, 7]. Notably, several researchers have highlighted the importance of performing resistance training exercises with maximal effort (i.e., move the load as rapidly as possible) to improve sport-specific high-velocity strength [8, 14, 15]. On this basis, the bench press throw (BPT) with lighter loads (30–60% 1-RM) is often recommended for explosive power training because it allows the lifter to push through the entire ascending movement [12, 16]. Furthermore, previous studies show superior acceleration, muscle activity, force, velocity, power output, and improvements in throwing distance and velocity with the performance of BPT compared to the traditional bench press using loads lower than 70% 1-RM [10, 11, 13].

Nevertheless, one of the more difficult regions of the exercise is the transition from descending to ascending the barbell [17]. Typically, this region is often standardized so that the barbell should stop at or just above chest level to avoid the bounce effect [10]. The bounce is the result of letting the barbell collide with the chest which immediately creates a momentum to help accelerate the ascending barbell. BPT with the bounce technique (BPT$_{bounce}$) has been utilized in training by elite track and field throwing athletes, as it may provide a more explosive and specific bench press variation. Still, to the authors' best knowledge, no previous study has examined the chronic effects of the bounce technique in BPT. However, Krajewski and colleagues [18] compared the acute effects of performing the conventional deadlift with either the pause or bounce technique. Twenty resistance-trained men performed two sets of 5 repetitions at 75% of 1-RM with both techniques. The bounce technique reduced the force requirements and lift time in both the early phase (0.0–0.1s) of the lift and the entire ascending phase.

The effects of BPT$_{bounce}$ may be comparable to the drop-jump. A drop-jump may result in high level of force development prior to the ascending phase, through a stretch-shortening cycle type action that stores the elastic energy, triggering spinal reflexes as the muscles stretches, and thereby enhancing the potentiation due to the pre-stretched muscles to a greater extent than a countermovement jump [19–22]. Therefore, the aim of this study was to examine the effects of bench press throw with bounce (BPT$_{bounce}$) and a without bounce (BPT) on throwing velocity (penalty and 3-step), 1-repetition maximum (1-RM) in bench press, and average power output (20–60kg) in bench press among handball players. It was hypothesized that the BPT$_{bounce}$ training group would increase throwing velocity and power more than the BPT group, while the BPT group would increase in the 1-RM more than the BPT$_{bounce}$ group.

## Methods

### Design

The study used a within- and between groups design in which the subjects were randomized to train twice per week with either BPT or BPT$_{bounce}$ for 8 weeks in addition to regular team

handball training. Test variables pre-and post-intervention consisted of throwing velocity (7m penalty and 3-step), bench press 1-RM, and average power output profile (20-60kg) in bench press.

## Participants

Subjects were recruited from two different handball teams and randomized into the two training groups BPT and $BPT_{bounce}$. Of note, each team was randomized into the two groups meaning that each team had equal number of participants in each group. This was done to counteract the possible bias due to the training routines and training/testing equipment. Initially, 19 amateur handball players were recruited; however, two players were injured (not related to the intervention), and one did not attend the post-test. Sixteen subjects completed at least 12 training sessions and were included in the data analysis. The average training attendance was 16 ($\pm$ 2.0 sessions) and 16 ($\pm$ 2.7 sessions) for the BPT group and $BPT_{bounce}$ group. Details of the subjects are included in Table 1.

## Ethics statement

All subjects were informed with written and verbal instructions regarding the implications and potential side effects of participating in this experiment. The study was conducted from October to December 2020, and the present procedures were performed in accordance with the Declaration of Helsinki and approved by the Norwegian Centre for Research Data (ref. 288211).

## Procedures

Four to five days before the pre-test, a familiarization with BPT and $BPT_{bounce}$ technique was performed. The familiarization session aimed to familiarize the subjects with both the BPT and $BPT_{bounce}$ technique by completing several attempts with each technique on loads between 30–60% of self-reported 1-RM (e.g., 20–70 kg). Of note, the handball players were experienced with resistance training and were tested in bench press 1-RM frequently (e.g., 3–5 times) each year. One-to-two repetitions for each load were conducted with no more than five loads for each technique. In the BPT technique, the subjects were instructed to lower the barbell and lightly touch (no bounce allowed) the chest (sternum position) and immediately press upwards with maximal voluntary intent until projecting the barbell (i.e., throw the barbell). Similar instructions were given for the $BPT_{bounce}$ technique, except the instruction to bounce the barbell off the chest. For both techniques, subjects were instructed with the following statement: "the goal is to generate as high velocity during the ascending phase as possible, with a fast but controlled lowering velocity". In the BPT, trials were omitted if the barbell bounced or if the

**Table 1. The subjects' characteristics.**

| Group | BPT | $BPT_{bounce}$ |
|---|---|---|
| Age (yr) | 17.9 ± 3.9 | 20.4 ± 5.0 |
| Body mass (kg) | 72.5 ± 9.2 | 71.8 ± 8.5 |
| Height (cm) | 183 ± 6.5 | 178 ± 6.7 |
| RT experience (yr) | 2.3 ± 3.3 | 1.6 ± 1.7 |
| Handball experience (yr) | 7.1 ± 3.2 | 9.2 ± 4.4 |

No significant differences were observed between the groups at pre-test (p $\geq$ 0.158). BPT = bench press throw; $BPT_{bounce}$ = bench press throw with bounce, RT = Resistance training.

descending phase was terminated before touching the sternum lightly. For the BPT$_{bounce}$, trials were rejected if the bar did not clearly bounce off the chest. In both techniques, trials were rejected if the hip lifted from the bench or if any hesitation occurred in the transition from the descending to the ascending phase.

Before each test session, the subjects were instructed to complete a 5-minute general warm-up on a treadmill or stationary bike. The warm-up continued in the lab with dynamic stretches for the pectoralis, anterior deltoid, and triceps brachii, followed by 10 repetitions in the bench press with 20kg, 4 repetitions at 50% of self-reported 1-RM and 2 repetitions with 75% of self-reported 1-RM. Preferred grip width and back position on the bench were measured and controlled before each lift. All testing and training were conducted in Smith machines.

The pre- and post-tests consisted of two testing days. On the first day, the load-power test and 1-RM, test was conducted. In the load-power test, the subjects performed 1–3 attempts separated with 2–3 minutes rest at loads increasing from 20–60kg with both BPT techniques performed in a randomized order. To assess the power output, a linear encoder (Ergotest Innovation, Stathelle, Norway) was attached to the barbell. The linear encoder measured barbell vertical displacement and time with a resolution of 0.019 mm and a sampling rate of 200 Hz. Using the commercial software Musclelab v.10 (Ergotest Innovation, Stathelle, Norway), the average power (e.g., from the lowest to the highest barbell position) was calculated from the ascending phase of the BPT for each load. The attempts with the highest average power were selected for the analysis.

The second day of testing was conducted 5–7 days after the first test and consisted of measuring maximal throwing velocity in a penalty throw (7m) and a 3-step handball throw. After a general warm-up, both tests followed a procedure inspired by Saeterbakken et al. [5]. Groups of three subjects tested in rotation with a 60-second rest between attempts, performing 5–10 maximal throws. The test was terminated when the velocity decreased after the 5$^{th}$ attempt. The penalty shot was performed behind the 7m line and followed regular penalty rules, with the front foot on the ground during the throw. The 3-step throw was performed behind the 9m dotted line, and subjects were allowed a 3-step run-up. Subjects were instructed to throw the ball (mass 480 g, circumference 58 cm) as fast and straightforward as possible [5]. Maximal ball velocity was measured with a Stalker Radar gun (The Stalker ATS II; Radar Sales, Plymouth, MN, USA) with an accuracy of ± 3%. The radar was located 1 meter behind the participant at ball height during the throw [23]. The average of the three best throws was used in further analysis [5]. The test-retest coefficient of variation (CV) for the three best throwing velocities used in the analyses were 1.49 and 1.23 for the penalty and 3-step throw.

**Training programs.**   Due to national traveling restrictions following the Covid-19 pandemic, only six subjects (three subjects in each group) from one of the recruited handball teams were supervised every session. The other ten subjects were supervised in the first two sessions and one session midway through the intervention. The unsupervised subjects were instructed to train in pairs and to encourage each other to perform each lift with maximal voluntary effort and with a proper bounce (e.g., a significant and visible compression of the chest). A researcher had weekly contact with them, and all subjects delivered a training log for each week. Both groups received the same power training program and were asked to continue their regular team handball training. In addition, the subjects were encouraged to continue their usual resistance training routines, but refrain from additional resistance training involving the chest, shoulder, and triceps muscles.

Each training session was initiated with a warm-up and included: dynamic stretches for the pectoralis major, anterior deltoid, and triceps brachii muscle, 10 repetitions with an unloaded bar (20kg), 6 repetitions with 50% of that session's training load and 4 repetitions with 70% of that session's training load. The training program (Table 2) was based on previous

**Table 2. Details of the 8 weeks power training program.**

| Week | Weekly sessions | Resistance | Sets | Repetitions | Rest between sets |
|------|-----------------|------------|------|-------------|-------------------|
| 1 | 2 | 40% of 1-RM | 3 | 5 | 3 minutes |
| 2 | 2 | 50% of 1-RM | 3 | 4 | 3 minutes |
| 3 | 2 | 60% of 1-RM | 3 | 3 | 3 minutes |
| 4 | 2 | 40% of 1-RM | 3 | 5 | 3 minutes |
| 5 | 2 | 50% of 1-RM | 3 | 4 | 3 minutes |
| 6 | 2 | 60% of 1-RM | 3 | 3 | 3 minutes |
| 7 | 2 | 40% of 1-RM | 3 | 5 | 3 minutes |
| 8 | 2 | 50% of 1-RM | 3 | 4 | 3 minutes |

recommendations for power training [12, 24] and a recent bench throw study [25]. The 1-RM result from the pre-test was used to calculate training load in weeks 1–4, whereas a new 1-RM test (identical procedures as described previously) was conducted after week 4 of the intervention to adjust the loading.

During the intervention, the subjects reported their weekly numbers of team handball sessions and resistance training sessions targeting the upper body. The post-test was performed 6–8 days after the intervention to maximize adaptations to the training intervention while minimizing fatigue.

**Statistics.** All baseline variables were tested for normality (Shapiro Wilk) and visually inspected. To examine potential differences in team handball training, resistance training or change in relative resistance in the loads for the power test, independent T-tests were used. Split-plot ANOVA (within-subject factor: time (pre and post); between-subject factor: BPT-technique (BPT and $BPT_{bounce}$)) was used to determine the effects of the intervention on average power output, maximal strength, 7m- and 3-step throwing velocity. Magnitude of the effects was determined using Cohen's d. An effect size of $< 0.2$ was considered trivial, 0.2–0.5 small, 0.5–0.8 medium and $> 0.8$ large [26]. The significance level was set to $\leq 0.05$ and all data are reported as mean ± standard deviation (SD) if nothing else is stated.

## Results

There was no significant difference between the two groups in weekly team handball ($p = 0.387$) or resistance training sessions ($p = 0.109$) during the intervention. The BPT reported 2.6 ± 1.7 and 3.3 ± 1.0 team handball and resistance training sessions per week whereas the $BPT_{bounce}$ reported 1.9 ± 1.2 and 2.6 ± 0.5 team handball and resistance training sessions per week. Furthermore, no significant differences in subject anthropometrics ($p \geq 0.158$), maximal ball velocity ($p \geq 0.246$) and 1-RM ($p = 0.629$) were observed at pre-test.

### Throwing velocity

No significant interaction ($F = 0.08$–$2.407$, $p = 0.539$–$0.929$) or significant main effect for group ($F = 1.058$–$1.1225$, $p = 0.290$–$0.324$) was observed for the 7m and 3-step throwing velocity, but a main effect for time ($F = 50.120$–$53.185$, $p < 0.001$–$0.005$) was observed (Figs 1 and 2, and S1 Table). Post hoc test demonstrated a 5.1% (84.70 ± 3.87 km/h vs. 89.00 ± 5.49 km/h, $p = 0.016$, ES = 0.91) and 2.3% non-significant (92.75 ± 5.36 km/h vs. 94.92 ± 5.49 km/h, $p = 0.114$, ES = 0.40) improvement for the 7m and 3-step throwing velocity for the $BPT_{bounce}$ group. For the BPT, a 5.2% (81.15 ± 7.41 km/h vs. 85.35 ± 7.29 km/h, $p < 0.001$, ES = 0.57) and 3.8% (88.24 ± 9.13 km/h vs. 91.59 ± 8.37 km/h, $p = 0.048$, ES = 0.38) improvement was observed for the 7m and 3-step throwing velocity, respectively.

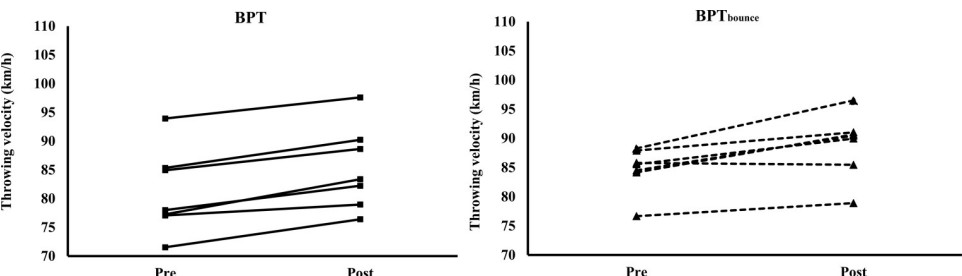

**Fig 1. The individual throwing velocity (km/h) before (pre) and after (post) the bench press throw intervention from the penalty position for the BPT and BPT_bounce group.**

## 1-RM

No significant interaction (F = 0.110, p = 0.746) or significant main effect for group (F = 0.137, p = 0.718) was observed, but a significant main effect for time (F = 35.670, p < 0.001) was observed for the 1-RM test. Post hoc tests demonstrated an 8.7% (64.64 ± 17.17 kg vs. 75.71 ± 19.83, p < 0.001, ES = 0.60) and 10.3% (66.07 ± 13.06 kg vs. 72.86 ± 14.61 kg, p < 0.001, ES = 0.49) increase for the BPT_bounce and the BPT group respectively.

## Power output in bench press throw

When measuring power output during bench press throw with bonce, no significant interaction (F = 0.066–2.477, p = 0.142–0.802) or significant main effect for group (F = 0.453–1.467, p = 0.254–0.513) or time (F = 1.763–4.389, p = 0.060–0.209) was observed with exception of significant main effect for time using the 50kg load (F = 9.780, p = 0.011). All post hoc tests and details are presented in Table 3.

For the bench press throw without bounce, no significant interaction (F = 0.003–2.989, p = 0.109–0.957) or significant main effect for group (F = 0.239–0.407, p = 0.217–0.634) was observed across the loads without bouncing the barbell. A significant main effect for time was observed for 20kg, 50kg and 60kg (F = 7.953–18.512, p = 0.002–0.015), but not 30kg and 40kg (F = 4.224–4.435, p = 0.057–0.062). All post hoc tests and details are presented in Table 3.

The bench press loads 20kg, 30kg, 40kg, 50kg and 60kg represented in the pre-test 31–78% of the 1-RM load for the BPT_bounce group and 31–87% for the BPT group. At post-test, the loads represented 28–76% of the 1-RM load for the BPT_bounce group and 28–85% for the BPT group. No significant differences of the loads (e.g., percent of 1-RM) were observed between the groups at pre-test (p = 0.334–0.940) or post-test (p = 0.449–0.993).

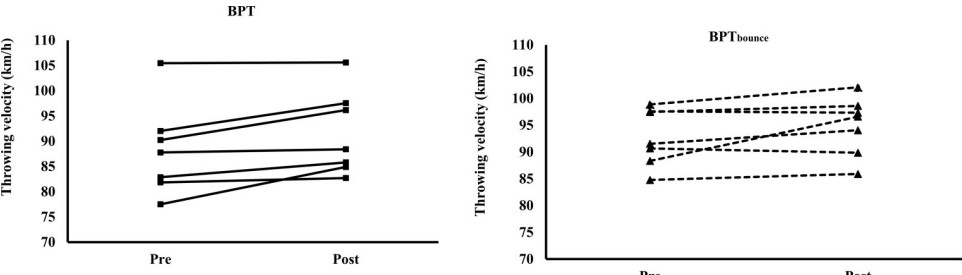

**Fig 2. The individual throwing velocity (km/h) before (pre) and after (post) the bench press throw intervention from the 3-step handball throw for the BPT and BPT_bounce group.**

**Table 3. Changes in average power output (W) in bench press for bench press throw and bench press throw with bounce.**

| | BPT group* | | | | | BPT$_{bounce}$ group* | | | | |
|---|---|---|---|---|---|---|---|---|---|---|
| | Pre | Post | % change | p-value | ES* | Pre | Post | % change | p-value | ES |
| Power (W) in BPT | | | | | | | | | | |
| 20 kg | 280 ± 63 | 306 ± 54 | 9.1 | 0.018* | 0.43 | 307 ± 56 | 316 ± 76 | 3.1 | 0.806 | 0.14 |
| 30 kg | 315 ± 86 | 355 ± 71 | 12.7 | 0.048* | 0.51 | 362 ± 72 | 365 ± 106 | 0.9 | No ME | 0.03 |
| 40 kg | 307 ± 119 | 359 ± 90 | 17.0 | No ME* | 0.50 | 359 ± 92 | 365 ± 143 | 1.6 | No ME | 0.04 |
| 50 kg | 300 ± 122 | 347 ± 108 | 15.9 | 0.056 | 0.69 | 356 ± 78 | 397 ± 100 | 11.6 | 0.060 | 0.45 |
| 60 kg | 275 ± 134 | 332 ± 133 | 20.9 | 0.184 | 0.43 | 317 ± 65 | 376 ± 111 | 18.8 | 0.164 | 0.65 |
| Power (W) in BPT$_{bounce}$ | | | | | | | | | | |
| 20 kg | 287 ± 64 | 321 ± 56 | 11.7 | No ME | 0.56 | 323 ± 49 | 328 ± 77 | 1.4 | No ME | 0.07 |
| 30 kg | 332 ± 96 | 370 ± 84 | 11.4 | No ME | 0.42 | 385 ± 85 | 391 ± 117 | 1.3 | No ME | 0.05 |
| 40 kg | 325 ± 142 | 373 ± 115 | 14.7 | No ME | 0.37 | 410 ± 101 | 408 ± 152 | 0.5 | No ME | -0.01 |
| 50 kg | 327 ± 125 | 356 ± 138 | 8.9 | 0.028* | 0.22 | 408 ± 91 | 443 ± 128 | 8.5 | 0.252 | 0.31 |
| 60 kg | 304 ± 161 | 356 ± 196 | 17.2 | No ME | 0.29 | 386 ± 92 | 423 ± 148 | 9.8 | No ME | 0.30 |

*Significant difference between pre- and posttest (p < 0.05). BPT = Bench press throw, BPT$_{bounce}$ = Bench press throw with bounce, ES = Effect size, ME = main effect.

## Discussion

The purpose of this study was to investigate the training effects of bench press throw with or without the bounce technique. The main findings were that eight weeks of power training with BPT or BPT $_{bounce}$ had similar effects on throwing velocity, maximal strength, and power output in amateur handball players.

In contrast to the hypotheses, similar effect of the two techniques were observed. This may be due to similar stimuli for adaptation as both techniques were trained with maximum voluntary effort and the intent to develop force as fast as possible throughout the entire ascending movement of the barbell. Furthermore, the resistance training protocol had similar training volume and similarities in techniques (e.g., targeting same muscle groups), and intensities (e.g., % of 1-RM). This may explain the similar results. In addition, the rate of muscular tension development and motor unit activation may have been relatively similar independent of the techniques [21, 27] as both groups performed BPT. Therefore, the difference between the techniques in the present study might have been insignificant to evoke different responses. Still, the possibility that some players performed a greater bounce than other cannot be rejected. A small bounce would make the training intervention close to identical and could potentially explain the findings. Importantly, the test leader attended as many training sessions as possible to promote maximal effort and a proper visual bounce. Furthermore, both groups trained with the same volume and load (sets x repetitions x load) suggesting that the workload between the groups was the same. Previous studies have compared different workloads such as heavy resistance training (>70% of 1-RM) with ballistic power training (< 30% of 1-RM) and reported similar improvement in sprint, jump height, and throwing performance [28, 29]. Still, to exploit the elastic energy and stretch-shortening cycle from the descending phase to the ascending phase with maximal acceleration of the loads, a considerable requirement of muscle power and force are required [30, 31], especially for the BPT$_{bounce}$ group. Based on the BPTbounce groups'relative 1-RM strength level (1-RM/body weight = 0.98), it is plausible that their strength level was too low to exploit the bounce effect maximally in BPT. For example, higher drop jump heights (>60cm) have demonstrated lower reactive strength index than lower heights [32]. Reactive strength index is calculated by dividing the jump height by ground contact time and has proven reliable and a useful tool to measure the ability to quickly change

from eccentric to concentric muscle action [33, 34]. In the aforementioned study [32], the ground contact time was longer using higher drop jump heights (>60cm) which means that the ability to rapidly absorb and then transmit the energy to a propulsive contraction decreased with increasing drop jump heights. Still, the relative lower limb strength was not included in the study [32], which could support our speculation that the subjects in the present study were not strong enough to exploit the bounce effect. Furthermore, non-professional volleyball players have demonstrated greater effects in different jump types during six weeks of countermovement jump training than drop jump training (17 vs. 7%) [35]. These findings could implicate that the subjects in the present study may not have been trained specifically for eccentric strength, which can result in reduced ability to absorb and transmit the energy to a concentric movement [36]. Importantly, maximal effort and the intention to develop force rapidly have previously been accounted as a critical stimulus for improving high-velocity performance in resistance training [37, 38]. For example, Sakamoto and colleagues [10] showed that bench throw training (30–50% RM) significantly increased throwing distance and maximum strength compared to no significant increase with the traditional bench press technique (e.g., no barbell throw).

The intervention may have provided a too low stimuli due to a combination of low loads, reduction in handball matches, duration of the intervention, or total training volume to detect differences between the groups in throwing velocity. The subjects of this study completed an average of 16 intervention sessions (i.e., six weekly sets) over eight weeks and the resistance training program was designed as a supplement to the subjects' team handball training. However, no physical contact or matches were allowed during the intervention due to the national and local COVID-19 regulation which could potentially have influenced the findings with reduced handball session intensity and throwing training (e.g., low loads training with maximal velocity). It could be speculated that a longer training period may have affected the muscular action more (i.e., neurological and morphological) and improved motor coordination, leading to greater effect of the bounce technique. However, this is speculative and cannot be answered by the present study's recordings. Nevertheless, the consistent performance of the sport-specific skill in conjunction with resistance training might be pivotal to transfer effects of from the resistance training to throwing velocity [8, 14, 15]. An increased training volume and an extended intervention period could have allowed potential differences between the groups [39]. Still, the increase in maximal strength for the two groups (8.7% and 10.2%) was similar to the 10% increase reported by Sakamoto et al. [10] who completed a 12 week intervention with two sessions per week.

Both groups improved the 7m penalty throw, whereas only the BPT improved the 3-step throwing velocity. This may be the result of the penalty throw being better at isolating improvements to the upper body musculature, as the 3-step involves more complex motor skills. Nevertheless, the effects on 7m throwing velocity in this study are similar to the 2% improvement reported by McEvoy and Newton [11] who incorporated a similar intervention concurrently with regular sport practice. The present study population may not be optimal for such a specific investigation. Amateur athletes may respond to a broad range of training stimuli and is typically less sensitive to the specifics of training [7, 15, 28]. For athletes with only a few years of resistance- or handball experience, additional throwing and general resistance training results in positive outcomes [7]. The speculation is supported by two reviews on throwing velocity who both stated that there is no definitive answer to which type of training which produces the greatest increase in throwing velocity [4, 7].

Regarding the bench press 1-RM results, it was hypothesized that the BPT would increase 1-RM to a greater degree than $BPT_{bounce}$. This hypothesis was based on the principle of specificity and that the 1-RM test was carried out with a technique more similar (no bounce

allowed) to the training of the BPT group. Contrary to the hypothesis, results showed similar improvements between the groups. Similar training volume, intensity, and workloads, in addition to low relative bench press strength (0.94 and 0.98 at baseline), are most likely the explanation of the findings. Different findings have been reported in a study comparing heavy resistance training and ballistic power training [28, 29], but not in all [40]. Importantly, power training with maximal effort can improve 1-RM strength [2].

Despite no significant differences between the groups in power output across loads, the BPT improved power output only at the 20kg and 30kg loads (9.1% and 12.7%, respectively) whereas BPT$_{bounce}$ demonstrated no significant differences across the loads. These loads were lower loads than the one being used in the intervention (Table 2) and thereby not according to the load/movement velocity specific response typically reported elsewhere [2, 12, 38]. Typically, it is considered easier to increase strength than velocity, especially when initial strength is low [16, 41]. This may explain the improvement in 1-RM strength even though no significant differences were observed between the groups. Furthermore, Cuevas-Aburto et al. [40] demonstrated similar increase in bench press 1-RM comparing a strength-oriented training program to a ballistic training program. Importantly, an increase in subjects' force-generating capacity increases their potential to become faster at any given force or resistance [42], potentially explaining the increased throwing velocity.

The present study has several limitations which the reader needs to be mindful of. First, the study suffered from a relatively small number of subjects and type 2 error cannot be ruled out when comparing the techniques. Using the post-test results from the 7m throwing test to calculate the minimal sample size to detect significant difference ($\alpha$ level of 0.05, and $\beta$ level of 80%), 12 subjects in each group was required. Additionally, and based on the post-test results, the statistical power in the present study was 44%. Also, the difference between the techniques might be greater with heavier loads (>60% 1RM). However, we designed a power training program using loads between 40–60% of 1-RM. Still, whether greater resistance training experience (e.g., strong athletes) or heavier loads might be more beneficial for one of the bench press throw techniques is beyond the scope of the present study, but should be examined in further studies. If introducing the bounce technique with a heavier relative load, it should be considered against the possibility of increased injury risk, especially if the subjects are inexperienced. Also, due to low access to appropriate subjects (handball players) during the COVID-19 pandemic, the study could not include a control group. Researchers may have these considerations in mind when interpreting the results or planning to investigate the effects of bounce in the future.

In conclusion, the results demonstrated similar effects of the BPT and BPT$_{bounce}$ techniques on maximal throwing velocity, maximal strength, and power output in amateur handball players. For athletes where the strength component is less developed, the present results indicated that the bounce technique is not of significant importance. Importantly, the findings of this study must be interpreted in the context of both techniques being performed as explosively as possible. It is important to note that the findings are limited to short-term power training among amateur athletes. In that case, more strength-oriented training could possibly render similar improvements to throwing velocity while increasing maximal strength to a larger degree, which is considered a critical long-term adaptation for athletes involved in explosive endeavors. However, once adequate strength has been developed, the use of more explosive and specific training variations is considered increasingly important [12, 15]. Yet, this possibility requires further study.

## Supporting information

**S1 Table. Data overview.**
(PDF)

## Acknowledgments

We thank the participants for their enthusiastic participation in this project. There are no conflict of interest.

## Author Contributions

**Conceptualization:** Jørund Løken, Tom Erik Jorung Solstad, Vidar Andersen, Atle Hole Saeterbakken.

**Data curation:** Jørund Løken.

**Formal analysis:** Jørund Løken, Atle Hole Saeterbakken.

**Investigation:** Vidar Andersen, Atle Hole Saeterbakken.

**Methodology:** Jørund Løken, Tom Erik Jorung Solstad, Nicolay Stien, Vidar Andersen.

**Project administration:** Atle Hole Saeterbakken.

**Resources:** Tom Erik Jorung Solstad, Nicolay Stien, Vidar Andersen.

**Supervision:** Tom Erik Jorung Solstad, Atle Hole Saeterbakken.

**Validation:** Vidar Andersen.

**Writing – original draft:** Jørund Løken, Atle Hole Saeterbakken.

**Writing – review & editing:** Tom Erik Jorung Solstad, Nicolay Stien, Vidar Andersen.

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
