## [Decision Letter · Decision Letter 0]

19 Oct 2021

PONE-D-21-31850Effects of bouncing the barbell in bench press on throwing velocity and strength among handball playersPLOS ONE

Dear Dr. Saeterbakken,

Thank you for submitting your manuscript to PLOS ONE. After careful consideration, we feel that it has merit but does not fully meet PLOS ONE’s publication criteria as it currently stands. Therefore, we invite you to submit a revised version of the manuscript that addresses the points raised during the review process.

Please, address point-to-point all reviewers' issues.

We look forward to receiving your revised manuscript.

Kind regards,

Luca Paolo Ardigò, Ph.D.

Academic Editor

PLOS ONE

Journal Requirements:

When submitting your revision, we need you to address these additional requirements. 1. Please ensure that your manuscript meets PLOS ONE's style requirements, including those for file naming. The PLOS ONE style templates can be found at https://journals.plos.org/plosone/s/file?id=wjVg/PLOSOne_formatting_sample_main_body.pdf and https://journals.plos.org/plosone/s/file?id=ba62/PLOSOne_formatting_sample_title_authors_affiliations.pdf  2. Please provide additional details regarding participant consent. In the ethics statement in the Methods and online submission information, please ensure that you have specified (1) whether consent was informed and (2) what type you obtained (for instance, written or verbal, and if verbal, how it was documented and witnessed). If your study included minors, state whether you obtained consent from parents or guardians. If the need for consent was waived by the ethics committee, please include this information. 3. Thank you for stating the following financial disclosure:  "The funders had no role in study design, data collection and analysis, decision to publish, or preparation of the manuscript." At this time, please address the following queries: a) Please clarify the sources of funding (financial or material support) for your study. List the grants or organizations that supported your study, including funding received from your institution. b) State what role the funders took in the study. If the funders had no role in your study, please state: “The funders had no role in study design, data collection and analysis, decision to publish, or preparation of the manuscript.”c) If any authors received a salary from any of your funders, please state which authors and which funders.d) If you did not receive any funding for this study, please state: “The authors received no specific funding for this work.” Please include your amended statements within your cover letter; we will change the online submission form on your behalf.

Additional Editor Comments:

Please, address point-to-point all reviewers' issues.

Reviewers' comments:

Reviewer's Responses to Questions

**Comments to the Author**

1. Is the manuscript technically sound, and do the data support the conclusions?

Reviewer #1: Yes

Reviewer #2: Yes

2. Has the statistical analysis been performed appropriately and rigorously? 

Reviewer #1: Yes

Reviewer #2: Yes

3. Have the authors made all data underlying the findings in their manuscript fully available?

Reviewer #1: Yes

Reviewer #2: Yes

4. Is the manuscript presented in an intelligible fashion and written in standard English?

Reviewer #1: Yes

Reviewer #2: Yes

5. Review Comments to the Author

Reviewer #1: Abstract

Line 21: Remove the bracket.

Introduction

Line 77-78: I would suggest adding the term stretch-shortening cycle to this sentence such as for example: “A drop-jump may result in high level of force prior to the ascending phase, WITH A STRETCH-SHORTENING CYCLE TYPE ACTION THAT STORES the elastic energy, …”

Methods

Participants

Since you were using a sample of convenience (two handball teams), then you probably did not calculate a statistical power analysis to determine the appropriate number of subjects to achieve sufficient statistical power. Please calculate a post-test statistical power analysis so the reader knows the extent of statistical power of this study’s measures.

Line 119-120: You might want to remind the readers that the handball players were experienced with resistance training and were familiar with their 1RM and thus the self-reported 1RM should have been accurate.

Line 125-131: How did you determine or ensure a similar bounce. Although you mention lowering the bar in a controlled manner, it is possible that some individuals would incorporate a far greater bounce than others. Would this not insert a high degree of variability into the training routine?

Line 138-142: How was power monitored? Was there a video analysis to monitor bar velocity or was there a line attached from the barbell to a device to monitor velocity? Table 3 illustrates the Watts but I do not see or must have missed where you explained how watts were calculated.

Line 162: unsupervised is one word, no hyphen necessary.

Line 281-284: I would add an explanation to this information that the individuals in that study may not have had the relative lower body strength to rapidly absorb or stretch the musculotendinous unit and then transition to a propulsive contraction. This would be in alignment with your previous statement that your subjects may not have been strong enough. In addition, it could be speculated that if your subjects had not trained specifically for eccentric strength then perhaps there might have been a specific eccentric strength deficit (i.e., training specificity). With a lack of prior eccentric training, individuals can have a lack of balance between concentric and eccentric strength.

Line 336: I have never seen this word before: contrastive; and I have been speaking only English for 64 years. Please change to “In contrast,…”.

Line 331-350: The power output differences were not significant as reported in the results (no interaction (F = 0.066 – 2.477, p = 0.142 – 0.802) or main effect for group (F = 0.453 – 1.467, p = 0.254 – 0.513)). This paragraph makes the reader think that something significant did happen. Without significance then the variability was greater than the mean difference and thus essentially there is no real difference. For example, the authors state: “As power is the product of both force and velocity, a possible explanation of the findings could be that the BPT emphasized the subjects’ ability to generate force more than the BPTbounce.” But if there is no significance then there are no sig different findings and this there is no need for an explanation of these non-significant findings. Please remove this paragraph or change the rationale and message of the paragraph to reflect the lack of significant difference.

Reviewer #2: This is a well written, reasoned, conducted and analysed study. The hypotheses are clear, the findings are as well, the interpretation is sound and based on the results and appropriate literature, there is no interpretative over-reach, and the limitations are stated. This is a tight study that adds to the body of literature on sport specific power training and i have only minor comments below. Well done.

Abstract

Instead of stating “no difference” please state no significant difference if it can fit into the abstract word limit.

Introduction

Line 67-68: please change “creates a momentum to help accelerating” to “creates momentum to help accelerate”

Procedures

Line 125 please change “barbell in the chest.” to “barbell off the chest.”

Results

In all cases where you state “no difference” or “no interaction” or similar please change to “no significant difference” or “no significant interaction”

Discussion

Line 291 please change “no increase” to “no significant increase”

6. PLOS authors have the option to publish the peer review history of their article (what does this mean?). If published, this will include your full peer review and any attached files.

Reviewer #1: **Yes: **David Behm

Reviewer #2: **Yes: **Eric Helms

---

## [Author Response · Author response to Decision Letter 0]

2 Nov 2021

We would like to thank the reviewers for your valuable comments and suggestions. We believe they have contributed to improve the quality of the manuscript. Under you will find our point-to-point answers and we have made changes in the manuscript accordingly.

Reviewer #1: Abstract

Line 21: Remove the bracket.

On behalf of all authors, we apologize for this typo. The bracket has been removed. 

Introduction

Line 77-78: I would suggest adding the term stretch-shortening cycle to this sentence such as for example: “A drop-jump may result in high level of force prior to the ascending phase, WITH A STRETCH-SHORTENING CYCLE TYPE ACTION THAT STORES the elastic energy, …”

Solid clarification of our point. The sentence has been re-written accordingly. Thank you. 

Methods

Participants

Since you were using a sample of convenience (two handball teams), then you probably did not calculate a statistical power analysis to determine the appropriate number of subjects to achieve sufficient statistical power. Please calculate a post-test statistical power analysis so the reader knows the extent of statistical power of this study’s measures.

Based on the 7m throwing velocity, we have calculated the statistical power and the sample size needed to detect significant difference between the groups. This has been added in the discussion (ln. 365 – 368) to elaborate on our speculations and limitations of the present study. 

Briefly, the statistical power was 44% and a minimum of 12 subjects was required in each training group. 

Line 119-120: You might want to remind the readers that the handball players were experienced with resistance training and were familiar with their 1RM and thus the self-reported 1RM should have been accurate.

Good point. The sentence “Of note, the handball players were experienced with resistance training and were tested in 1-RM frequently (e.g., 3 – 5 times) each year” has been added to the manuscript. 

Line 125-131: How did you determine or ensure a similar bounce. Although you mention lowering the bar in a controlled manner, it is possible that some individuals would incorporate a far greater bounce than others. Would this not insert a high degree of variability into the training routine?

It`s good question and difficult to provide an 100% accurate answer as we did not include measurement of the vertical descending displacement for each subject. Importantly, the instructions were “the goal is to generate as high velocity during the ascending phase as possible, with a fast but controlled lowering velocity”. Furthermore, we were more interested in ecological perspective and the “proof of concept”. That being said, we cannot reject the hypotheses of individual differences which has been added as a possible explanation in the discussion (ln 280 -284). 

By “control” we referred to “not dropping the barbell from the starting position”. One familiarization session was conducted before the pre-test to ensure proper bouncing. Furthermore, one test leader attended as many training sessions as possible given the Covid-19 regulations to make sure a proper bounce was conducted. That being said, we cannot reject your hypotheses that some subjects (greater chest volume, less fear, and greater arching) may generated a far greater bounce. 

In the pilot test using the same instructions and loads, we measured a 3.6 - 4.7cm longer vertical displacement in the descending phase. Typically, greater loads resulted in less bounce (probably as result of pain and fear). Furthermore, the training load was 40 – 60% of 1-RM (See table 2). We therefore hypothesized that the loading would result in different amount of bounce, but that during the training intervention, the subjects would conduct more similar bouncing across the loads. 

In the pilot testing, we also included a metronome to control the lowering speed, but experienced that the subjects` attention was towards the metronome and not to conduct a bounce or a maximal explosive bench press throw. 

Of note, our research group are examining your hypotheses, but not on an individual level but the descending instruction. Using the peak power loads, the subjects (resistance trained men) are being asked to lower the barbell slow, medium, and as fast as possible with and without the bounce. We passed 20 subjects yesterday and are aiming to reach 30. That being said, from these data we could calculate individual variation of the amount of bounce using the three lower velocities. 

In the manuscript, we have added some sentence in the methods (ln 170-171) and discussion (ln 280-284). Thanks. 

Line 138-142: How was power monitored? Was there a video analysis to monitor bar velocity or was there a line attached from the barbell to a device to monitor velocity? Table 3 illustrates the Watts but I do not see or must have missed where you explained how watts were calculated.

We apologize for this mistake and acknowledge your observant reading skills. We used a linear encoder attached to the barbell to measure the average velocity (e.g., from the start of the ascending phase to barbell`s highest position). Please see ln. 143 – 148 for further details. 

Line 162: unsupervised is one word, no hyphen necessary.

Thank you, changed accordingly. 

Line 281-284: I would add an explanation to this information that the individuals in that study may not have had the relative lower body strength to rapidly absorb or stretch the musculotendinous unit and then transition to a propulsive contraction. This would be in alignment with your previous statement that your subjects may not have been strong enough. In addition, it could be speculated that if your subjects had not trained specifically for eccentric strength then perhaps there might have been a specific eccentric strength deficit (i.e., training specificity). With a lack of prior eccentric training, individuals can have a lack of balance between concentric and eccentric strength.

Added, as suggested (ln. 297 – 306). 

Line 336: I have never seen this word before: contrastive; and I have been speaking only English for 64 years. Please change to “In contrast,…”.

Google translate has… 😊. Changed to “In contrast, … and later the sentence was deleted….

Line 331-350: The power output differences were not significant as reported in the results (no interaction (F = 0.066 – 2.477, p = 0.142 – 0.802) or main effect for group (F = 0.453 – 1.467, p = 0.254 – 0.513)). This paragraph makes the reader think that something significant did happen. Without significance then the variability was greater than the mean difference and thus essentially there is no real difference. For example, the authors state: “As power is the product of both force and velocity, a possible explanation of the findings could be that the BPT emphasized the subjects’ ability to generate force more than the BPTbounce.” But if there is no significance then there are no sig different findings and this there is no need for an explanation of these non-significant findings. Please remove this paragraph or change the rationale and message of the paragraph to reflect the lack of significant difference.

Thank you for bringing this to our attention. Yes, there were no interactions or main effects for group, but a main effect for time for some of the loads. Importantly, post hoc tests have only been conducted within a group (pre – post) and not between groups. However, your example “as power is the product of…” we acknowledge that the paragraph needs to be re-written and clarified. Therefore, we have re-written the paragraph, removed major parts of it, and clarified the findings using words as “no significant difference”. Please ln 351 – 362. 

Thank you for taking the time and sharing your knowledge in this review proses. Your comments and suggestions will be included and kept in mind in future papers. Thank you! 

 

Reviewer #2: This is a well written, reasoned, conducted and analysed study. The hypotheses are clear, the findings are as well, the interpretation is sound and based on the results and appropriate literature, there is no interpretative over-reach, and the limitations are stated. This is a tight study that adds to the body of literature on sport specific power training and i have only minor comments below. Well done.

Thank you so much!

Abstract

Instead of stating “no difference” please state no significant difference if it can fit into the abstract word limit.

We managed to add “no significant difference” in abstract. Good point, added as suggested. 

Introduction

Line 67-68: please change “creates a momentum to help accelerating” to “creates momentum to help accelerate”

Changed, thank you

Procedures

Line 125 please change “barbell in the chest.” to “barbell off the chest.”

Changed, thank you

Results

In all cases where you state “no difference” or “no interaction” or similar please change to “no significant difference” or “no significant interaction”

Added, as suggested. 

Discussion

Line 291 please change “no increase” to “no significant increase”

Added, as suggested. 

Thank you for taking the time and sharing your knowledge in this review proses. Your comments and suggestions will be included and kept in mind in future papers. Thank you! 

---

## [Decision Letter · Decision Letter 1]

8 Nov 2021

Effects of bouncing the barbell in bench press on throwing velocity and strength among handball players

PONE-D-21-31850R1

Dear Dr. Saeterbakken,

We’re pleased to inform you that your manuscript has been judged scientifically suitable for publication and will be formally accepted for publication once it meets all outstanding technical requirements.

Kind regards,

Luca Paolo Ardigò, Ph.D.

Academic Editor

PLOS ONE

Additional Editor Comments (optional):

Congratulations for the interesting work.

Reviewers' comments:

Reviewer's Responses to Questions

**Comments to the Author**

1. If the authors have adequately addressed your comments raised in a previous round of review and you feel that this manuscript is now acceptable for publication, you may indicate that here to bypass the “Comments to the Author” section, enter your conflict of interest statement in the “Confidential to Editor” section, and submit your "Accept" recommendation.

Reviewer #1: All comments have been addressed

Reviewer #2: All comments have been addressed

2. Is the manuscript technically sound, and do the data support the conclusions?

Reviewer #1: Yes

Reviewer #2: (No Response)

3. Has the statistical analysis been performed appropriately and rigorously? 

Reviewer #1: Yes

Reviewer #2: (No Response)

4. Have the authors made all data underlying the findings in their manuscript fully available?

Reviewer #1: Yes

Reviewer #2: (No Response)

5. Is the manuscript presented in an intelligible fashion and written in standard English?

Reviewer #1: Yes

Reviewer #2: (No Response)

6. Review Comments to the Author

Reviewer #1: Authors have adequately addressed my concerns.

The web site indicates that I need a minimum of 80 characters. This additional explanatory sentence should reach that count or else I could go blah, blah, blah, blah, blah etc.

Reviewer #2: (No Response)

7. PLOS authors have the option to publish the peer review history of their article (what does this mean?). If published, this will include your full peer review and any attached files.

Reviewer #1: **Yes: **David G. Behm

Reviewer #2: **Yes: **Eric Helms

---

## [Editor Report · Acceptance letter]

10 Nov 2021

PONE-D-21-31850R1 

Effects of bouncing the barbell in bench press on throwing velocity and strength among handball players 

Dear Dr. Saeterbakken:

I'm pleased to inform you that your manuscript has been deemed suitable for publication in PLOS ONE. Congratulations! Your manuscript is now with our production department. 

Kind regards, 

on behalf of

Dr. Luca Paolo Ardigò 

Academic Editor

PLOS ONE